# Cultural, sociopolitical, environmental and built assets supporting health and well-being in Torres Strait Island communities: protocol for a scoping review

Torres Webb,[1] Kathryn Meldrum ,[1] Melissa Kilburn,[1] Valda Wallace,[1] Sarah Russell,[1] Rachel Quigley,[1] Edward Strivens[1,2]

TW and KM are joint first authors.

¹College of Medicine and Dentistry, James Cook University, Cairns, Queensland, Australia
²Cairns and Hinterland Hospital and Health Service, Queensland Health, Cairns, Queensland, Australia

**Correspondence to**
Dr Kathryn Meldrum;
kathryn.meldrum@jcu.edu.au

## ABSTRACT

**Introduction** Risks to an individual's health should be considered alongside the environmental, sociocultural and sociopolitical context(s) in which they live. Environmental mapping is an approach to identifying enablers and barriers to health within a community. The Indigenous Indicator Classification System (IICS) framework has been used to map the environment in Australian Indigenous communities. The IICS is a four-level nested hierarchical framework with subject groups including culture, sociopolitical and built at the top of the hierarchy and indicators at the bottom. The objective of this scoping review is to map the cultural, sociopolitical, environmental and built assets that support health and well-being that exist in each Torres Strait Island community.

**Methods and analysis** This review will be conducted according the Joanna Briggs Institute (JBI) method for scoping reviews. It will include sources that identify cultural, sociopolitical, environmental and built assets that support health and well-being that exist in each Torres Strait Island community. Databases to be searched include: Informit; Scopus; Web of Science; HealthInfoNet, BioOne Complete and Green File. Sources of unpublished and grey literature will be located using Google and Google Scholar. Searches will be limited to the English language and literature published since January 2018 to ensure that the assets mapped reflect current conditions on each island. Data that answers the research question will be extracted from sources and recorded in an adaptation of the IICS. Quantitative analysis of the data will include summing each asset for individual islands and their associated clusters. Data will be presented graphically, diagrammatically, or in tabular form depending on what approach best conveys its meaning.

**Ethics and dissemination** The Far North Queensland Human Research Ethics Committee (reference HREC/2022/QCH/88 155-1624) has approved this study. Dissemination of the review's findings will be led by Torres Strait Islander members of the research team through conferences and peer-reviewed publications.

## INTRODUCTION

The WHO estimates that 24% of all global deaths are linked with the environment.[1]

## STRENGTHS AND LIMITATIONS OF THIS STUDY

⇒ The registration and publication of the protocol insures transparency of the review process.
⇒ The scoping review will be conducted according to the method prescribed by the Joanna Briggs Institute.
⇒ The Indigenous Indicator Classification System used to identify and classify assets has been validated.
⇒ Data collection will be limited to sources that are able to be located through desk-based searches which may mean that some cultural, sociopolitical, environmental and built assets will not be recorded.

Consequently, understanding the interactions between the environment and health, especially those exacerbated by climate change, can assist with managing their impact. Mapping the environment is an approach to identifying barriers to health and well-being within a community.[2–4] Environmental mapping is a robust approach because, as Rose[5] identified, the risk to an individual's health should be considered alongside the environmental, sociocultural and sociopolitical context(s) in which they live. One approach to environmental mapping in an Australian Indigenous community in Western Australia focused mainly on barriers to health attributable to the environment.[2] This focus failed to recognise that the environment can also be a protective factor,[6] especially in Australian Indigenous communities.[3 7 8] Le Gal *et al*[9] supported a broader conceptualisation of health, inclusive of the environment, for Indigenous Australians. The rationale for broader conceptualisation builds on the previous work of Daniel *et al*.[3] Their review[3] argued that the context and composition of the environment act together as risks for

health and well-being in Australian Indigenous communities. The authors defined context as the 'properties of places' (p330) and composition as 'collective attributes of people in a place' (p330).[3] Environmental context and composition coalesce to interact with an individual's genetic, behavioural and stress axes to predispose them to chronic disease.

### The Indigenous Indicator Classification System as an approach to environmental mapping

Early work by Marks *et al*[10 11] proposed the Indigenous Indicator Classification System (IICS) as a framework that could be used to map the 'distinct historical, social and cultural contexts of Indigenous communities' (p93). The IICS was developed in 2006 with input from Indigenous peoples from Canada, Aotearoa/New Zealand and Australia[3] and validated in 2009.[11]

The IICS is a hierarchical four-level system with subject groups at the highest level. At the time of development, subject groups consisted of culture, sociodemographic, sociopolitical, socioeconomic, built, and natural environment, psychosocial and social organisations. For example, cultural assets include the background and history of peoples and the traditional activities and cultural responsibilities that they undertake.[10] Sociopolitical assets include, but are not limited to, community economic resources, Indigenous self-government and autonomy and labour market and working conditions whereas built assets include but are not limited to the natural environment, housing and transportation. The subject groups are the broadest classification level to which each indicator group or goal dimension is allocated to.[10] Domains are the broadest of the nested hierarchical categories[10] and identify the focus of the goal dimension and/or indicator group.[12] Goal dimensions describe the parameters for classifying indicators and could be used

to identify potential targets for health policies and/ or programmes.[10] Consequently, in the IICS, they often describe a desirable community outcome. For example, community business and economic development.[4] Indicators are measurable information[13] able to be found in documents, grey literature or through community conversations. In Australia, health indicators, such as those used in the IICS, describe particular elements of health or aspects of performance.[14] The link(s) between domains and indicators are important for explaining and providing a platform for investigating the complexity of influences on Indigenous peoples' health[10] and well-being. Figure 1 outlines the IICS. Figure 2 illustrates the IICS using two examples: the built and natural environment and sociopolitical subject groups.

### The context of the protocol and scoping review

The setting for this study is the Torres Strait (Zenadth Kes) situated between the tip of the eastern Australian mainland and Papua New Guinea. There are over 200 islands in the Torres Strait located in a vast geographical area of over $44\,000$ km$^2$. However, only 17 islands are permanently inhabited.[15] In 2016, approximately 7403 people lived on the islands of the Torres Strait, 86.6% of whom identified as Australian First Nations peoples.[16]

### Rationale and aims

The context of this scoping review is a broader project funded by the Australian National Health and Medical Research Council (grant number MRF2016931) designed to address significant gaps in existing knowledge and develop an understanding of key assets for a strengths-based approach to targeting diet and activity components of chronic disease risk for First Nations peoples living in the Torres Strait. The broader project is designed to:

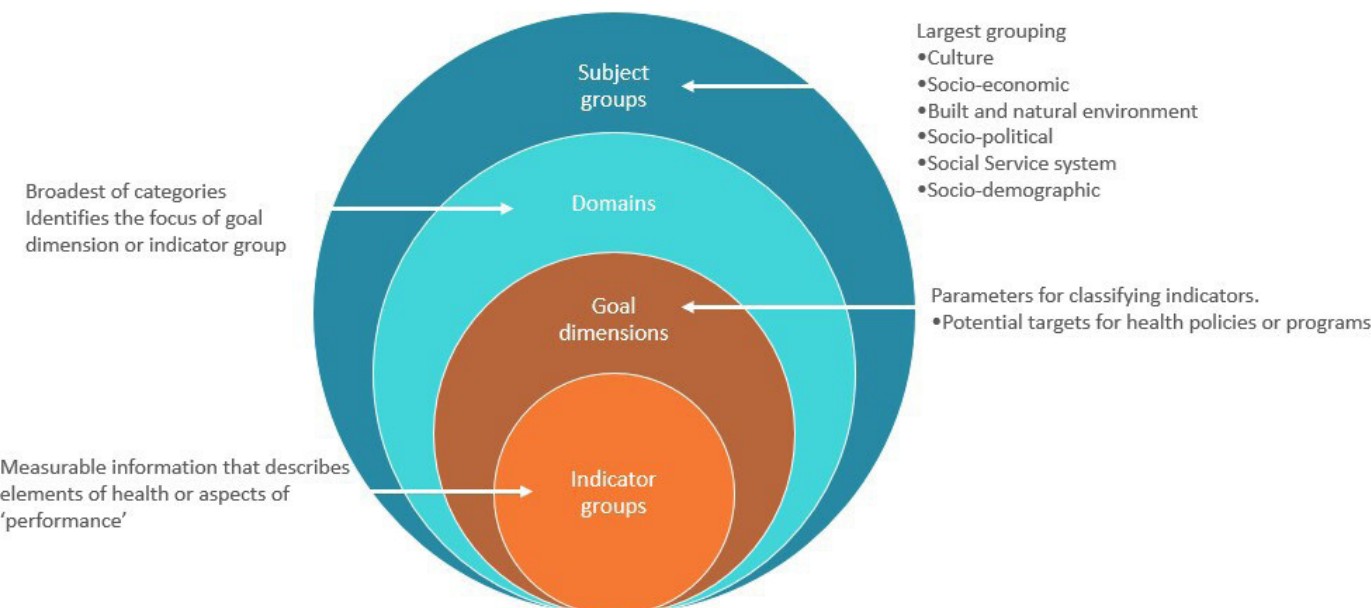

**Figure 1** The Indigenous Indicator Classification System (adapted from Ref 2).

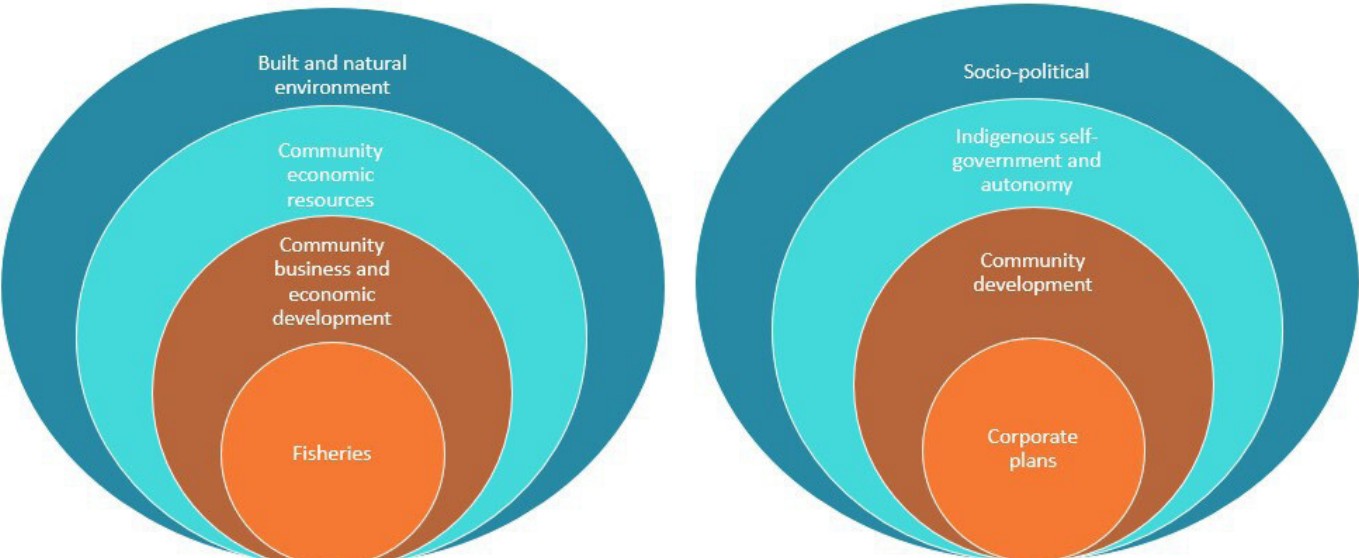

**Figure 2** Examples of the Indigenous Indicator Classification System.

► identify the strengths and barriers to good health in Torres Strait communities,

► identify dietary and activity behaviours of Australian First Nations peoples living in the Torres Strait and

► use this information to develop community-led chronic disease prevention interventions at an individual and service provider level.

The aim of the scoping review that this protocol refers to is to addresses dot point 1 (above) initially by mapping the cultural, sociopolitical, natural and built environment of each of the inhabited Islands of the Torres Strait. The research question guiding this scoping review is: what cultural, sociopolitical, environmental and built assets that support health and well-being exist in each Torres Strait Island community?

## METHODS AND ANALYSIS

The proposed scoping review will be conducted in accordance with the Joanna Briggs Institute (JBI) methodology for scoping reviews.[17] Both this protocol and subsequent scoping review will be reported using the Preferred Reporting Items for Systematic Reviews and Meta-analyses Extension for Scoping Review (PRISMA-ScR) guidelines.[18]

### Inclusion criteria

In line with the JBI methodology, the Participant, Concept, Context framework will be used to guide inclusion criteria and search strategy for the scoping review.

### Participants

The scoping review will include sources that identify and/or describe any cultural, sociopolitical, natural and built assets (hereafter referred to collectively as assets) that support health and well-being in Torres Strait Island communities. Assets may include people and programmes.

### Concept

The concepts to be examined in the scoping review will include assets that support health and well-being in Torres Strait Island communities. Specific data to be extracted includes:

► the island and/or community where the asset is located,

► the type of asset.

Sources excluded from the data set include any assets identified from a secondary source. For example, data in published literature must come from research conducted by the authors. The rationale for excluding secondary sources is that it may be out of date or may not be accurate. In addition, the information must describe actual assets not aspirational ones that may be detailed in regional plans.

### Context

Any source that details the assets of Torres Strait Island communities will be considered for inclusion in the scoping review. The sources may be obtained from database searches or from grey literature located on key regional governance websites such as Torres Strait Regional Authority, Torres Strait Island Regional Council, Gur A Baradharaw Kod Torres Strait Sea and Land Council, and Torres Shire Council.

### Types of sources

Relevant sources that have been peer reviewed or have been obtained from grey literature will be included in the data set. Sources may include annual reports, strategic, corporate, operational or implementation plans, evaluations, planning documents and published literature. Critical reviews, literature reviews and systematic reviews

will not be included in the data set because they are not primary data sources. However, their reference lists will be reviewed for relevant primary data sources.

Inclusion criteria in order of priority will be:
1. information that identifies assets available/present in a Torres Strait Island community,
2. uses primary data sources,
3. has been published since 2018.

The scoping review is registered with The Open Science Framework (https://osf.io/2wv8c).

## Search strategy

The search strategy will aim to locate published and unpublished sources. A three-step search strategy will be used in this review. First, an initial limited search of Web of Science and Informit was undertaken to identify relevant literature. The text words contained in the titles and abstracts of relevant sources, and the index terms used to describe them were used to develop a full search strategy tailored to each information source. For example, a pilot search strategy is illustrated in the online supplement materials (online supplemental appendix A). Step one was supported by a subject-specific liaison librarian. The reference lists of all included sources of evidence will be screened for additional sources. Additionally, reference lists of systematic reviews on the same or similar topic will be scrutinised for appropriate papers, reports or other data sources.

Published or unpublished reports or other data sources in the English language and available since the year 2018 to present will be included in the data set. Sources published since 2018 will be used because they are more likely to identify assets currently present in Torres Strait Island communities.

Databases to be searched include:
▶ Informit;
▶ Scopus;
▶ Web of Science;
▶ BioOne Complete;
▶ HealthInfoNet;
▶ Green File.

Unpublished and grey literature will be identified using Google, Google Scholar and website searches. Google and Google Scholar searches will use the search string Torres Strait Island* AND community AND environment AND plan OR Policy OR report with a date limiter from 1 January 2018—date of search.

## Source of evidence selection

At the commencement of step 2, all identified citations will be collated and uploaded into EndNote (V.20.1)[19] and data will be subsequently managed according to the method proposed by Peters.[20] A pilot test will then be conducted as described by Peters *et al*,[17] following which, titles and abstracts will be screened by two independent reviewers for assessment against the inclusion criteria for the review. Potentially relevant sources will be retrieved in full and subsequently assessed against the inclusion

criteria by two independent reviewers. Reasons for exclusion of full-text sources of evidence that do not meet the inclusion criteria will be recorded and reported in the scoping review. Any disagreements that arise between the reviewers at each stage of the selection process will be resolved through discussion. Where agreement cannot be achieved through discussion, an additional reviewer/s will review the source. The results of the search and the study inclusion process will be reported in full in the final scoping review and presented in a PRISMA-ScR flow diagram.[21]

## Data extraction

Before data extraction, the IICS will be adapted for the context of this study by the co-first authors (TW, KM). Subsequently, data will be manually extracted from sources included in the scoping review by two independent reviewers using a database adaptation of the IICS.[4 10 11] Data extracted from sources will include specific information about the participants, concept and context. Any further adaptations to the IICS made during data extraction will be detailed in the scoping review. Any disagreements will be managed in the manner previously discussed. If appropriate, source authors will be contacted to request missing or additional information, where required. In addition, key informants in regional governance bodies will be contacted to confirm source information when necessary.

## Data analysis and presentation

Data will be analysed quantitatively to identify how it is distributed geographically across the Torres Strait Islands. Assets used will be categorised into the overarching groups:
▶ Cultural.
▶ Sociopolitical.
▶ Environmental.
▶ Built.

Once an asset has been categorised into a group, it will subsequently be allocated into an indicator group by referring to the relevant indicator statement. The presence of an asset will be recorded by typing a '1' into the relevant column in the database. Once data collection is complete, asset sums will be calculated for each indicator, for each Island and Island group.

Data will be presented graphically, diagrammatically or in tabular form depending on what approach that best represents the data. For example, using Geographic Information System mapping software may be used to indicate what assets are available on each island as this approach potentially conveys greater meaning than tabulating it. A narrative summary will accompany the tabulated and/or charted results and will describe how the results relate to the review's objective and question.

## Patient and public involvement

As previously indicated, this study is phase one of a broader four phase study aimed at establishing an evidence-base

for codesigned interventions aimed at reducing chronic disease risk for First Nations peoples living on the islands of the Torres Strait. The broader project is situated within, and contributes to, a project already underway focusing on healthy ageing in the Torres Strait. Findings from this project will be incorporated into the wider Framework for Healthy Ageing being developed for the region. The research team also works with an existing Knowledge Circle (Indigenous Reference Group) that oversees the team's research in the region. The Knowledge Circle comprises Aboriginal and Torres Strait Islander community members, aged care workers, healthcare staff and academics who are committed to supporting the health of people in their communities. Projects are also conducted in partnership with local health and aged care providers in the Torres Strait.

## ETHICS AND DISSEMINATION

This scoping review will map the assets that support health and well-being of Torres Strait Islanders living on the inhabited Islands of the Torres Strait. At a broader level, the findings will be of interest to Torres Strait Islanders and policy-makers. They could potentially be used to advocate for more and/or appropriate assets to support health and well-being across the Torres Strait. At the level of the broader project, the findings of the scoping review will be used to support the second phase. The second phase of the project is to yarn with community members about strengths and barriers to health and well-being that are part of their lives living on a Torres Strait Island. Phase 2 of this study will be led by Torres Strait Islander members of the research team with findings disseminated using a range of strategies including ongoing community engagement, conference presentations and publications.

Finally, as this is a desk-based review of publicly available sources, ethical clearance is not necessary for this study. However, ethical clearance for the broader project has been obtained from the Far North Queensland Human Research Ethics Committee (reference HREC/2022/QCH/88155-1624).

**Acknowledgements** Janet Catterall Subject-specific Liaison Librarian.

**Contributors** TW and KM designed the protocol and drafted the manuscript. All authors reviewed and approved the final manuscript.

**Funding** This work was supported by an Australian National Health and Medical Research Council grant number [MRF2016931].

**Competing interests** None declared.

**Patient and public involvement** Patients and/or the public were involved in the design, or conduct, or reporting, or dissemination plans of this research. Refer to the Methods section for further details.

**Patient consent for publication** Not applicable.

**Provenance and peer review** Not commissioned; externally peer reviewed.

**ORCID iD**
Kathryn Meldrum http://orcid.org/0000-0003-1846-1596

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
