## [Reviewer comments · BMJ Open]

ARTICLE DETAILS

TITLE (PROVISIONAL)	Cultural, socio-political, environmental, and built assets supporting health and wellbeing in Torres Strait Island communities: Protocol for a scoping review
AUTHORS	Webb, Torres; Meldrum, Kathryn; Kilburn, Melissa; Wallace, Valda; Russell, Sarah; Quigley, Rachel; Strivens, Edward

VERSION 1 – REVIEW

REVIEWER	Hemez Ange Aurélien Kouassi Institut International d'Ingénierie de l'Eau et de l'Environnement
REVIEW RETURNED	27-Aug-2023

GENERAL COMMENTS	Thanks to the authors for the work and to the editor for allowing me to review this paper. But personally, I don't find the work relevant to the scientific community. It's too community-oriented and will probably be suitable for a local journal. The method is not sufficiently clear and looks like a mixture of several methodologies, including PRISMA. There's a lot of ambiguity, and even the way it's written, or even the way some of the authors are cited, isn't right. I didn't see any results, discussion or even a conclusion to the study. The introduction is a little too light. I was expecting to see a map as the authors were talking about mapping, but I was left wanting more. Personally, I'm not in favor of publishing this paper, at least as long as it's in its current format. It would need to be greatly improved before it could be published. Also, the study area is not presented, the characteristics of the community studied and many other things are missing from this study.
---

REVIEWER	Julaine Allan Charles Sturt University
REVIEW RETURNED	20-Sep-2023

GENERAL COMMENTS	Thank you for the opportunity to review Cultural, socio-political, environmental, and built assets supporting health and wellbeing in Torres Strait Island communities: Protocol for a scoping review. This is a novel approach to assessing health and well-being in Australian Indigenous communities looking at both risk and protective factors in the environment. Page 5 line 4-5 needs a reference The description of the IICS is well done but it is complex. Fig. 1 is a very important illustration but is it possible to have more detail in the figure e.g. two subject groups to illustrate the system? All of them would be good but perhaps that requires too much detail?
--

	P.8 were any sources found in the initial search testing the search criteria? I would expect the grey literature to be a better source of information for this topic than the peer reviewed databases. More detail on the strategy for the grey lit search would be helpful. P.9 how does the data extraction tool adapt the IICS? Have you tried Covidence for reviewing - fantastic software especially for multiple reviewers Overall very well written manuscript, protocol clearly described and logical. Excellent introduction to the project.
--	--

VERSION 1 – AUTHOR RESPONSE

Reviewer 1

Comment	How addressed	Page
1. Work not relevant to the scientific community. It is too community-orientated and more suitable for a local journal.	Thank you for your perspective on the relevance of this protocol to the wider scientific community. We acknowledge that the context for our work is relatively small on a global scale. However, our approach to utilising an indicator scale that accounts for Indigenous peoples' holistic conceptualisation of the interaction between the environment and their health and wellbeing may be of interest to other researchers globally. In addition, it may also be of interest to researchers working with Indigenous communities globally who are often more significantly impacted by the effects of climate change.	n/a
2. Method not sufficiently clear and looks like a mixture of several methodologies including PRISMA	Thank you for your comment. We have reviewed the Methods and Analysis section of the protocol for readability and made changes where necessary. As this manuscript describes the protocol for a systematic scoping review, we have integrated the JBI methodology for scoping reviews which includes the PRISMA guidelines and associated flow chart. To identify the assets in each of the island communities we have used a previously published framework, the Indigenous Indicator Classification System (1-5)	6-10
3. A lot of ambiguity and citation of authors	Thank you for this observation. We have reviewed author citations, and all now align with BMJs requirements for citations. We have reviewed and edited the text for possible ambiguity. However, without specific examples it is difficult to address your perception of ambiguity.	3-11
4. No results, discussion or conclusion to the study,	The manuscript describes a protocol for a systematic scoping review. It does not describe the findings of a study and therefore does not contain results, a discussion and conclusion of a research study. We have included additional introductory sentences to	3

	situate the study within the global context.	
5. Expecting to see a map.	Mapping is the outcome of the systematic scoping review. Consequently, a map, if relevant will be provided when the findings of the scoping review are published.	n/a

Reviewer 2

Comment	How addressed	Page
6. Page 5 line 4-5 needs a reference	Thank you for this comment. We have cited Marks and colleagues (1) here.	5
7. Figure 1 needs more detail. For example, two subject groups to illustrate the system	Thank you for your suggestion. We appreciate that the IICS is complex. Consequently, we have included an additional diagram. Figure 1 outlines the IICs. Figure 2 provides two examples.	5
8. Were there any sources found in the initial search testing the search criteria? More detail on the strategy for the grey literature search would be helpful.	Thank you for this query. Yes, 75 sources were located using the initial search strategy using the Informit database. More information on the search strategy for Google and Google Scholar has been provided.	8
9. How does the data extraction tool adapt the IICS?	Thank you for this question. The IICS was adapted by the co-first authors before data searches were conducted. The adapted IICS was then converted into an Excel database. We have amended the text to clarify this. Also, this will be a manual not automated process, so we did not use Covidence. We appreciate your suggestion though.	9

VERSION 2 – REVIEW

REVIEWER	Julaine Allan Charles Sturt University
REVIEW RETURNED	18-Nov-2023

GENERAL COMMENTS	Thanks for making the changes The manuscript is much improved
--